# Discretely Relaxing Continuous Variables for tractable Variational Inference

**Trefor W. Evans**
University of Toronto
trefor.evans@mail.utoronto.ca

**Prasanth B. Nair**
University of Toronto
pbn@utias.utoronto.ca

## Abstract

We explore a new research direction in Bayesian variational inference with discrete latent variable priors where we exploit Kronecker matrix algebra for efficient and exact computations of the evidence lower bound (ELBO). The proposed "DIRECT" approach has several advantages over its predecessors; (i) it can exactly compute ELBO gradients (i.e. unbiased, zero-variance gradient estimates), eliminating the need for high-variance stochastic gradient estimators and enabling the use of quasi-Newton optimization methods; (ii) its training complexity is *independent* of the number of training points, permitting inference on large datasets; and (iii) its posterior samples consist of sparse and low-precision quantized integers which permit fast inference on hardware limited devices. In addition, our DIRECT models can exactly compute statistical moments of the parameterized predictive posterior without relying on Monte Carlo sampling. The DIRECT approach is not practical for all likelihoods, however, we identify a popular model structure which is practical, and demonstrate accurate inference using latent variables discretized as extremely low-precision 4-bit quantized integers. While the ELBO computations considered in the numerical studies require over $10^{2352}$ log-likelihood evaluations, we train on datasets with over two-million points in just seconds.

## 1 Introduction

Hardware restrictions posed by mobile devices make Bayesian inference particularly ill-suited for on-board machine learning. This is unfortunate since the safety afforded by Bayesian statistics is extremely valuable in many prominent mobile applications. For example, the cost of erroneous decisions are very high in autonomous driving or mobile robotic control. The robustness and uncertainty quantification provided by Bayesian inference is therefore extremely valuable for these applications provided inference can be performed on-board in real-time [1, 2].

Outside of mobile applications, resource efficiency is still an important concern. For example, deployed models making billions of predictions per day can incur substantial energy costs, making energy efficiency an important consideration in modern machine learning architectures [3].

We approach the problem of efficient Bayesian inference by considering discrete latent variable models such that posterior samples of the variables will be quantized and sparse, leading to efficient inference computations with respect to energy, memory and computational requirements. Training a model with a discrete prior is typically very slow and expensive, requiring the use of high variance Monte Carlo gradient estimators to learn the variational distribution. The main contribution of this work is the development of a method to rapidly learn the variational distribution for such a model without the use of any stochastic estimators; the objective function will be computed exactly at each iteration. To our knowledge, such an approach has not been taken for variational inference of large-scale probabilistic models.

In this paper, we compare our work not only to competing stochastic variational inference (SVI) methods for discrete latent variables, but also to the more general SVI methods for continuous latent variables. We make this comparison with continuous variables by discretely relaxing continuous priors using a discrete prior with a finite support set that contains much of the structure and information as its continuous analogue. Using this discretized prior we show that we can make use of Kronecker matrix algebra for efficient and exact ELBO computations. We will call our technique DIRECT (DIscrete RElaxation of ConTinous variables). We summarize our main contributions below:

- We efficiently and exactly compute the ELBO using a discrete prior even when this computation requires more likelihood evaluations than the number of atoms in the known universe. This achieves unbiased, zero-variance gradients which we show outperforms competing Monte Carlo sampling alternatives that give high-variance gradient estimates while learning.

- Complexity of our ELBO computations are *independent* of the quantity of training data using the DIRECT method, making the proposed approach amenable to big data applications.

- At inference time, we can exactly compute the statistical moments of the parameterized predictive posterior distribution, unlike competing techniques which rely on Monte Carlo sampling.

- Using a discrete prior, our models admit sparse posterior samples that can be represented as quantized integer values to enable efficient inference, particularly on hardware limited devices.

- We present the DIRECT approach for generalized linear models and deep Bayesian neural networks for regression, and discuss approximations that allow extensions to many other models.

- Our empirical studies demonstrate superior performance relative to competing SVI methods on problems with as many as 2 million training points.

The paper will proceed as follows; section 2 contains a background on variational inference and poses the learning problem to be addressed while section 3 outlines the central ideas of the DIRECT method, demonstrating the approach on several popular probabilistic models. Section 4 discusses limitations of the proposed approach and outlines some work-arounds, for instance, we discuss how to go beyond mean-field variational inference. We empirically demonstrate our approaches in section 5, and conclude in section 6. Our full code is available at `https://github.com/treforevans/direct`.

## 2   Variational Inference Background

We begin with a review of variational inference, a method for approximating probability densities in Bayesian statistics [4–9]. We introduce a regression problem for motivation; given $\mathbf{X} \in \mathbb{R}^{n \times d}$, $\mathbf{y} \in \mathbb{R}^n$, a $d$-dimensional dataset of size $n$, we wish to evaluate $y_*$ at an untried point $\mathbf{x}_*$ by constructing a statistical model that depends on the $b$ latent variables in the vector $\mathbf{w} \in \mathbb{R}^b$. After specifying a prior over the latent variables, $\Pr(\mathbf{w})$, and selecting a probabilistic model structure that admits the likelihood $\Pr(\mathbf{y}|\mathbf{w})$, we may proceed with Bayesian inference to determine the posterior $\Pr(\mathbf{w}|\mathbf{y})$ which generally requires analytically intractable computations.

Variational inference turns the task of computing a posterior into an optimization problem. By introducing a family of probability distributions $q_{\boldsymbol{\theta}}(\mathbf{w})$ parameterized by $\boldsymbol{\theta}$, we minimize the Kullback-Leibler divergence to the exact posterior [9]. This equates to maximization of the evidence lower bound (ELBO) which we can write as follows for a continuous or discrete prior, respectively

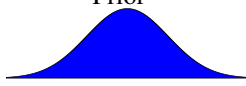

$$\mathsf{ELBO}(\boldsymbol{\theta}) = \int q_{\boldsymbol{\theta}}(\mathbf{w}) \Big( \log \Pr(\mathbf{y}|\mathbf{w}) + \log \Pr(\mathbf{w}) - \log q_{\boldsymbol{\theta}}(\mathbf{w}) \Big) d\mathbf{w}, \quad (1)$$

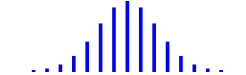

$$\mathsf{ELBO}(\boldsymbol{\theta}) = \mathbf{q}^T \Big( \log \boldsymbol{\ell} + \log \mathbf{p} - \log \mathbf{q} \Big), \quad (2)$$

where $\log \boldsymbol{\ell} = \{\log \Pr(\mathbf{y}|\mathbf{w}_i)\}_{i=1}^m$, $\log \mathbf{p} = \{\log \Pr(\mathbf{w}_i)\}_{i=1}^m$, $\mathbf{q} = \{q_{\boldsymbol{\theta}}(\mathbf{w}_i)\}_{i=1}^m$, and $\{\mathbf{w}_i\}_{i=1}^m = \mathbf{W} \in \mathbb{R}^{b \times m}$ is the entire support set of the discrete prior.

It is immediately evident that computing the ELBO is challenging when $b$ is large, since in the continuous case eq. (1) is a $b$-dimensional integral, and in the discrete case the size of the sum in eq. (2) generally increases exponentially with respect to $b$. Typically, the ELBO is not explicitly computed and instead, a Monte Carlo estimate of the gradient of the ELBO with respect to the variational

parameters $\boldsymbol{\theta}$ is found, allowing stochastic gradient descent to be performed. We will outline some existing techniques to estimate ELBO gradients with respect to the variational parameters, $\boldsymbol{\theta}$.

For continuous priors, the reparameterization trick [10] can be used to perform variational inference. The technique uses Monte Carlo estimates of the gradient of the evidence lower bound (ELBO) which is maximized during the training procedure. While this approach has been employed successfully for many large-scale models, we find that discretely relaxing continuous latent variable priors can improve training and inference performance when using our proposed DIRECT technique which computes the ELBO (and its gradients) exactly.

When the latent variable priors are discrete, reparameterization cannot be applied, however, the REINFORCE [11] estimator may be used to provide an unbiased estimate of the ELBO during training (alternatively called the score function estimator [12], or likelihood ratio estimator [13]). Empirically, the REINFORCE gradient estimator is found to give a high-variance when compared with reparameterization, leading to a slow learning process. Unsurprisingly, we find that our proposed DIRECT technique trains significantly faster than a model trained using a REINFORCE estimator.

Recent work in variational inference with discrete latent variables has largely focused on continuous relaxations of discrete variables such that reparameterization can be applied to reduce gradient variance compared to REINFORCE. One example is CONCRETE [14, 15] and its extensions [16, 17]. We consider an opposing direction by identifying how the ELBO (eq. (2)) can be computed exactly for a class of discretely relaxed probabilistic models such that the discrete latent variable model can be trained more easily then its continuous counterpart. We outline this approach in the following section.

## 3 DIRECT: Efficient ELBO Computations with Kronecker Matrix Algebra

We outline the central ideas of the DIRECT method and illustrate its application on several probabilistic models. The DIRECT method allows us to efficiently and exactly compute the ELBO which has several advantages over existing SVI techniques for discrete latent variable models such as, zero-variance gradient estimates, the ability to use a super-linearly convergent quasi-Newton optimizer (since our objective is deterministic), and the per-iteration complexity is independent of training set size. We will also discuss advantages at inference time such as the ability to exactly compute predictive posterior statistical moments, and to exploit sparse and low-precision posterior samples.

To begin, we consider a discrete prior over our latent variables whose support set $\mathbf{W}$ forms a Cartesian tensor product grid as most discrete priors do (e.g. any prior that factorizes between variables) so that we can write

$$\mathbf{W} = \begin{pmatrix} \bar{\mathbf{w}}_1^T & \otimes & \mathbf{1}_{\overline{m}}^T & \otimes & \cdots & \otimes & \mathbf{1}_{\overline{m}}^T \\ \mathbf{1}_{\overline{m}}^T & \otimes & \bar{\mathbf{w}}_2^T & \otimes & \cdots & \otimes & \mathbf{1}_{\overline{m}}^T \\ \vdots & & \vdots & & \ddots & & \vdots \\ \mathbf{1}_{\overline{m}}^T & \otimes & \mathbf{1}_{\overline{m}}^T & \otimes & \cdots & \otimes & \bar{\mathbf{w}}_b^T \end{pmatrix}, \tag{3}$$

where $\mathbf{1}_{\overline{m}} \in \mathbb{R}^{\overline{m}}$ denotes a vector of ones, $\bar{\mathbf{w}}_i \in \mathbb{R}^{\overline{m}}$ contains the $\bar{m}$ discrete values that the $i$th latent variable $w_i$ can take[1], $m = \bar{m}^b$, and $\otimes$ denotes the Kronecker product [18]. Since the number of columns of $\mathbf{W} \in \mathbb{R}^{b \times \overline{m}^b}$ increases exponentially with respect to $b$, it is evident that computing the ELBO in eq. (2) is typically intractable when $b$ is large. For instance, forming and storing the matrices involved naively require exponential time and memory. We can alleviate this concern if $\mathbf{q}$, $\log \mathbf{p}$, $\log \boldsymbol{\ell}$, and $\log \mathbf{q}$ can be written as a sum of Kronecker product vectors (i.e. $\sum_i \bigotimes_{j=1}^b \mathbf{f}_j^{(i)}$, where $\mathbf{f}_j^{(i)} \in \mathbb{R}^{\overline{m}}$). If we find this structure, then we never need to explicitly compute or store a vector of length $m$. This is because eq. (2) would simply require multiple inner products between Kronecker product vectors which the following result demonstrates can be computed extremely efficiently.

**Proposition 1.** *The inner product between two Kronecker product vectors* $\mathbf{k} = \otimes_{i=1}^b \mathbf{k}^{(i)}$, *and* $\mathbf{a} = \otimes_{i=1}^b \mathbf{a}^{(i)}$ *can be computed as follows [18],*

$$\mathbf{a}^T \mathbf{k} = \prod_{i=1}^b \mathbf{a}^{(i) \, T} \mathbf{k}^{(i)}, \tag{4}$$

*where $\mathbf{a}^{(i)} \in \mathbb{R}^{\overline{m}}$, $\mathbf{a} \in \mathbb{R}^{\overline{m}^b}$, $\mathbf{k}^{(i)} \in \mathbb{R}^{\overline{m}}$, and $\mathbf{k} \in \mathbb{R}^{\overline{m}^b}$.*

This result enables substantial savings in the computation of the ELBO since each inner product computation is reduced from the naive *exponential* $\mathcal{O}(\overline{m}^b)$ cost to a *linear* $\mathcal{O}(b\overline{m})$ cost.

We now discuss how the Kronecker product structure of the variables in eq. (2) can be achieved. Firstly, if the prior is chosen to factorize between latent variables, as it often is, (i.e. $\Pr(\mathbf{w}) = \prod_{i=1}^{b} \Pr(w_i)$) then $\mathbf{p} = \otimes_{i=1}^{b} \mathbf{p}_i$ admits a Kronecker product structure where $\mathbf{p}_i = \{\Pr(w_i=\bar{w}_{ij})\}_{j=1}^{\overline{m}} \in (0,1)^{\overline{m}}$. The following result demonstrates how this structure for $\mathbf{p}$ enables $\log \mathbf{p}$ to be written as a sum of $b$ Kronecker product vectors.

**Proposition 2.** *The element-wise logarithm of the Kronecker product vector $\mathbf{k} = \otimes_{i=1}^{b} \mathbf{k}^{(i)}$ can be written as a sum of $b$ Kronecker product vectors as follows,*

$$\log \mathbf{k} = \bigoplus_{i=1}^{b} \log \mathbf{k}^{(i)}, \tag{5}$$

*where $\mathbf{k}^{(i)} \in \mathbb{R}^{\overline{m}}$, $\mathbf{k} \in \mathbb{R}^{\overline{m}^b}$ contain positive values, and $\oplus$ is a generalization of the Kronecker sum [19] for vectors which we define as follows*

$$\bigoplus_{i=1}^{b} \log \mathbf{k}^{(i)} = \sum_{i=1}^{b} \left( \bigotimes_{j=1}^{i-1} \mathbf{1}_{\overline{m}} \right) \otimes \log \mathbf{k}^{(i)} \otimes \left( \bigotimes_{j=i+1}^{b} \mathbf{1}_{\overline{m}} \right). \tag{6}$$

The proof is trivial. We will first consider a mean-field variational distribution that factorizes over latent variables such that both $\mathbf{q} = \otimes_{i=1}^{b} \mathbf{q}_i$ and $\log \mathbf{q} = \oplus_{i=1}^{b} \log \mathbf{q}_i$ can be written as a sum of Kronecker product vectors, where $\mathbf{q}_j = \{\Pr(w_j=\bar{w}_{ji})\}_{i=1}^{\overline{m}} \in (0,1)^{\overline{m}}$ are used as the variational parameters, $\boldsymbol{\theta}$, with the use of the softmax function. For the mean-field case we can rewrite eq. (2) as

$$\mathsf{ELBO}(\boldsymbol{\theta}) = \mathbf{q}^T \log \boldsymbol{\ell} + \sum_{i=1}^{b} \mathbf{q}_i^T \log \mathbf{p}_i - \sum_{i=1}^{b} \mathbf{q}_i^T \log \mathbf{q}_i, \tag{7}$$

where we use the fact that $\mathbf{q}_i$ defines a valid probability distribution for the $i$th latent variable such that $\mathbf{q}_i^T \mathbf{1}_{\overline{m}} = 1$. We extend results to unfactorized prior and variational distributions later in section 4.

The structure of $\log \boldsymbol{\ell}$ depends on the probabilistic model used; in the worst case, $\log \boldsymbol{\ell}$ can always be represented as a sum of $m$ Kronecker product vectors. However, many models admit a far more compact structure where dramatic savings can be realized as we demonstrate in the following sections.

### 3.1 Generalized Linear Regression

We first focus on the popular class of Bayesian generalized linear models (GLMs) for regression. While the Bayesian integrals that arise in GLMs can be easily computed in the case of conjugate priors, for general priors inference is challenging.

This highly general model architecture has been applied in a vast array of application areas. Recently, Wilson et al. [20] used a scalable Bayesian generalized linear model with Gaussian priors on the output layer of deep neural network with notable empirical success. They also considered the ability to train the neural network simultaneously with the approximate Gaussian process which we also have the ability to do if a practitioner were to require such an architecture.

Consider the generalized linear regression model $\mathbf{y} = \boldsymbol{\Phi}\mathbf{w} + \boldsymbol{\epsilon}$, where $\boldsymbol{\epsilon} \sim \mathcal{N}(\mathbf{0}, \sigma^2 \mathbf{I})$, and $\boldsymbol{\Phi} = \{\phi_j(\mathbf{x}_i)\}_{i,j} \in \mathbb{R}^{n \times b}$ contains the evaluations of the basis functions on the training data. The following result demonstrates how the ELBO can be exactly and efficiently computed, assuming the factorized prior and variational distributions over $\mathbf{w}$ discussed earlier. Note that we also consider a prior over $\sigma^2$.

**Theorem 1.** *The ELBO can be exactly computed for a discretely relaxed regression GLM as follows*

$$\mathit{ELBO}(\boldsymbol{\theta}) = -\frac{n}{2}\mathbf{q}_\sigma^T \log \boldsymbol{\sigma}^2 - \frac{1}{2}\left(\mathbf{q}_\sigma^T \boldsymbol{\sigma}^{-2}\right)\left(\mathbf{y}^T\mathbf{y} - 2\mathbf{s}^T\left(\boldsymbol{\Phi}^T\mathbf{y}\right) + \mathbf{s}^T\boldsymbol{\Phi}^T\boldsymbol{\Phi}\mathbf{s} - \mathit{diag}(\boldsymbol{\Phi}^T\boldsymbol{\Phi})^T\mathbf{s}^2 +$$

$$\sum_{j=1}^{b} \mathbf{q}_j^T \mathbf{h}_j\right) + \sum_{i=1}^{b}\left(\mathbf{q}_i^T \log \mathbf{p}_i - \mathbf{q}_i^T \log \mathbf{q}_i\right) + \mathbf{q}_\sigma^T \log \mathbf{p}_\sigma - \mathbf{q}_\sigma^T \log \mathbf{q}_\sigma, \tag{8}$$

*where $\mathbf{q}_\sigma, \mathbf{p}_\sigma \in \mathbb{R}^{\overline{m}}$ are factorized variational and prior distributions over the Gaussian noise variance $\sigma^2$ for which we consider the discrete positive values $\boldsymbol{\sigma}^2 \in \mathbb{R}^{\overline{m}}$, respectively. Also, we use the shorthand notation $\mathbf{H} = \{\bar{\mathbf{w}}_j^2 \sum_{i=1}^n \phi_{ij}^2\}_{j=1}^b \in \mathbb{R}^{\overline{m} \times b}$, and $\mathbf{s} = \{\mathbf{q}_j^T \bar{\mathbf{w}}_j\}_{j=1}^b \in \mathbb{R}^b$.*

A proof is provided in appendix A of the supplementary material. We can pre-compute the terms $\mathbf{y}^T\mathbf{y}$, $\mathbf{\Phi}^T\mathbf{y}$, $\mathbf{H}$, and $\mathbf{\Phi}^T\mathbf{\Phi}$ before training begins (since these do not depend on the variational parameters) such that the final complexity of the proposed DIRECT method outlined in Theorem 1 is only $\mathcal{O}(b\overline{m} + b^2)$. This complexity is *independent* of the number of training points, making the proposed technique ideal for massive datasets. Also, each of the pre-computed terms can easily be updated as more data is observed making the techniques amenable to online learning applications.

**Predictive Posterior Computations**  Typically, the predictive posterior distribution is found by sampling the variational distribution at a large number of points and running the model forward for each sample. To exactly compute the statistical moments, a model would have to be run forward at every point in the hypothesis space with is typically intractable, however, we can exploit Kronecker matrix algebra to efficiently compute these moments exactly. For example, the exact predictive posterior mean for our generalized linear regression model is computed as follows

$$\mathbb{E}(y_*) = \sum_{i=1}^m q(\mathbf{w}_i) \int y_* \Pr(y_*|\mathbf{w}_i) dy_*, = \mathbf{\Phi}_* \mathbf{W} \mathbf{q} = \mathbf{\Phi}_* \mathbf{s}, \tag{9}$$

where $\mathbf{s} = \{\mathbf{q}_j^T \bar{\mathbf{w}}_j\}_{j=1}^b \in \mathbb{R}^b$, and $\mathbf{\Phi}_* \in \mathbb{R}^{1 \times b}$ contains the basis functions evaluated at $x_*$. This computation is highly efficient, requiring just $\mathcal{O}(b)$ time per test point. It can be shown that a similar scheme can be derived to exactly compute higher order statistical moments, such as the predictive posterior variance, for generalized linear regression models and other DIRECT models.

We have shown how to exactly compute statistical moments, and now we show how to exploit our discrete prior to compute predictive posterior samples extremely efficiently. This sampling approach may be preferable to the exact computation of statistical moments on hardware limited devices where we need to perform inference with extreme memory, energy and computational efficiency. The latent variable posterior samples $\widetilde{\mathbf{W}} \in \mathbb{R}^{b \times \text{num. samples}}$ will of course be represented as a low-precision quantized integer array because of the discrete support of the prior which enables extremely compact storage in memory. Much work has been done elsewhere in the machine learning community to quantize variables for storage compression purposes since memory is a very restrictive constraint on mobile devices [21–24]. However, we can go beyond this to additionally reduce computational and energy demands for the evaluation of $\mathbf{\Phi}_* \widetilde{\mathbf{W}}$. One approach is to constrain the elements of $\bar{\mathbf{w}}$ to be 0 or a power of 2 so that multiplication operations simply become efficient bit-shift operations [25–27]. An even more efficient approach is to employ basis functions with discrete outputs so that $\mathbf{\Phi}_*$ can also be represented as a low-precision quantized integer array. For example, a rounding operation could be applied to continuous basis functions. Provided that the quantization schemes are an affine mapping of integers to real numbers (i.e. the quantized values are evenly spaced), then inference can be conducted using extremely efficient integer arithmetic [28]. Either of these approaches enable extremely efficient on-device inference.

## 3.2   Deep Neural Networks for Regression

We consider the hierarchical model structure of a Bayesian deep neural network for regression. Considering a DIRECT approach for this architecture is not conceptually challenging so long as an appropriate neuron activation function is selected. We would like a non-linear activation that maintains a compact representation of the log-likelihood evaluated at every point in the hypothesis space, i.e. we would like $\log \ell$ to be represented as a sum of as few Kronecker product vectors as possible. Using a power function for the activation can maintain a compact representation; the natural choice being a quadratic activation function (i.e. output $x^2$ for input $x$).

It can be shown that the ELBO can be exactly computed in $\mathcal{O}(\ell\overline{m}(b/\ell)^{4\ell})$ for a deep Bayesian neural network with $\ell$ layers, where we assume a quadratic activation function and an equal distribution of discrete latent variables between network layers. This complexity evidently enables scalable Bayesian inference for models of moderate depth, and like we found for the regression GLM model of section 3.1, computational complexity is *independent* of the quantity of training data, making this approach ideal for large datasets. We outline this model and the computation of its ELBO in appendix D.

# 4   Limitations & Extensions

In generality, when the support of the prior is on a Cartesian grid, any prior, likelihood, or variational distribution (or log-distribution) can be expressed using the proposed Kronecker matrix representation, however, this representation will not always be compact enough to be practical. We can see this by viewing these probability distributions over the hypothesis space as high-dimensional tensors. In section 3, we exploited some popular models whose variational probability tensors, and whose prior, likelihood and variational log-probability tensors all admit a low-rank structure, however, other models may not admit this structure, in which case their representation will not be so compact. In the interest of generalizing the technique, we outline a likelihood, a prior, and a variational distribution that does not admit a compact representation of the ELBO and discuss several ways the DIRECT method can still be used to efficiently compute, or lower bound the ELBO. We hope that these extensions inspire future research directions in approximate Bayesian inference.

**Generalized Linear Logistic Regression**   Logistic regression models do not easily admit a compact representation for exact ELBO computations, however, we will demonstrate that we can efficiently compute a lower-bound of the ELBO by leveraging developed algebraic techniques. To demonstrate, we will consider a generalized linear logistic regression model which is commonly employed for classification problems. Such a model could easily be extended to a deep architecture following Bradshaw et al. [2], if desired.  All terms in the ELBO in eq. (7) can be computed exactly for this model except the term involving the log-likelihood, for which the following result demonstrates an efficient computation of the lower bound.

**Theorem 2.** *For a generalized linear logistic regression model with classification training labels* $\mathbf{y} \in \{0,1\}^n$, *the class-conditional probability* $\Pr(y_i{=}0|\mathbf{w}) = (1 + \exp(-\mathbf{\Phi}[i,:]\mathbf{w}))^{-1}$, *and with the assumption that training examples are sampled independently, the following inequality holds*

$$\mathbf{q}^T \log \boldsymbol{\ell} \geqslant -\mathbf{s}^T\left(\mathbf{\Phi}^T \mathbf{y}\right) - \sum_{i=1}^{n} \left\{ \begin{array}{ll} \prod_{j=1}^{b} \mathbf{q}_j^T \exp(-\phi_{ij}\bar{\mathbf{w}}_j) & \text{if } y_i = 0 \\ \prod_{j=1}^{b} \mathbf{q}_j^T \exp(\phi_{ij}\bar{\mathbf{w}}_j) - \sum_{j=1}^{b} \mathbf{q}_i^T \phi_{ij}\bar{\mathbf{w}}_j & \text{if } y_i = 1 \end{array} \right. \tag{10}$$

We prove this result in appendix B of the supplement. This computation can be performed in $\mathcal{O}(\bar{m}bn)$ time, where dependence on $n$ is evident unlike in the case of the exact computations described in section 3. As a result, stochastic optimization techniques should be considered. Using this lower bound, the log-likelihood is accurately approximated for hypotheses that correctly classify the training data, however, hypotheses that confidently misclassify training labels may be over-penalized. In appendix B we further discuss the accuracy of this approximation and discuss a stable implementation.

**Unfactorized Variational Distributions**   We now consider going beyond a mean-field variational distribution to account for correlations between latent variables. Considering a finite mixture of factorized categorical distributions as is used in latent structure analysis [29, 30], we can write $\mathbf{q} = \sum_{i=1}^{r} \alpha_i \bigotimes_{j=1}^{b} \mathbf{q}_j^{(i)}$, where $\boldsymbol{\alpha} \in (0,1)^r$ is a vector of mixture probabilities for $r$ components, and $\mathbf{q}_j^{(i)} = \{\Pr(w_j{=}\bar{w}_{jk}|i)\}_{k=1}^{\bar{m}} \in (0,1)^{\bar{m}}$.

While $\mathbf{q}$ can evidently be expressed as a compact sum of Kronecker product vectors, $\log \mathbf{q}$ is more challenging to compute than in the mean-field case, however, the following result demonstrates how we can lower-bound the term involving $\log \mathbf{q}$ in the ELBO (eq. (7)).

**Theorem 3.** *The following inequality holds when we consider a finite mixture of factorized categorical distributions for* $q_{\boldsymbol{\theta}}(\mathbf{w})$,

$$-\mathbf{q}^T \log \mathbf{q} \geqslant \max_{\{\mathbf{a}_i \in (0,1)^{\bar{m}}\}_{i=1}^{b}} 1 - \sum_{j=1}^{r} \alpha_j \left( \sum_{i=1}^{b} \mathbf{q}_i^{(j)\,T} \log \mathbf{a}_i + \alpha_j \prod_{i=1}^{b} \mathbf{q}_i^{(j)\,T} \frac{\mathbf{q}_i^{(j)}}{\mathbf{a}_i} + 2 \sum_{k=j+1}^{r} \alpha_k \prod_{i=1}^{b} \mathbf{q}_i^{(j)\,T} \frac{\mathbf{q}_i^{(k)}}{\mathbf{a}_i} \right),$$

*where* $\mathbf{a} = \otimes_{i=1}^{b} \mathbf{a}_i$, $\mathbf{a}_i \in (0,1)^{\bar{m}}$ *is the center of the Taylor series approximation of* $\log \mathbf{q}$.

We prove this result in appendix C and discuss a stable implementation. Note that if the mixture variational distribution $\mathbf{q}$ degenerates to a mean-field distribution equal to $\mathbf{a}$, then the ELBO will be computed exactly, and as $\mathbf{q}$ moves away from $\mathbf{a}$, the ELBO will be underestimated.

**Unfactorized Prior Distributions** To consider an unfactorized prior, we assume a prior mixture distribution given by $\mathbf{p} = \sum_{i=1}^{r} \alpha_i \bigotimes_{j=1}^{b} \mathbf{p}_j^{(i)}$. When we use this mixture distribution for the prior, $\mathbf{p}$ can evidently be expressed as a compact sum of Kronecker product vectors but $\log \mathbf{p}$ cannot. The following result demonstrates how we can still lower-bound the term involving $\log \mathbf{p}$ in the ELBO (eq. (2)). For simplicity, we assume that the variational distribution factorizes, however, the result could easily be extended to the case of a mixture variational distribution.

**Theorem 4.** *The following inequality holds when we consider a finite mixture of factorized categorical distributions for $p_{\boldsymbol{\theta}}(\mathbf{w})$,*

$$\mathbf{q}^T \log \mathbf{p} \geqslant \sum_{i=1}^{r} \alpha_i \sum_{j=1}^{b} \mathbf{q}_j^T \log \mathbf{p}_j^{(i)}$$

The proof is trivial by Jensen's inequality. Note that the equality only holds when the prior mixture degenerates to a factorized distribution with all mixture components equivalent.

**Unbiased Stochastic Entropy and Prior Expectation Gradients** We previously showed how to lower bound the ELBO terms $\mathbf{q}^T \log \mathbf{p}$ and $-\mathbf{q}^T \log \mathbf{q}$ when the variational and/or prior distributions do not factor, however, optimizing this bound introduces bias and does not guarantee convergence to a local optimum of the true ELBO. Here we reintroduce REINFORCE to deliver unbiased gradient estimates for these terms. The REINFORCE estimator typically has high variance, however, since gradient estimates for these terms are so cheap, a massive number of samples can be used per stochastic gradient descent (SGD) iteration to decrease variance. Since we can still compute the expensive $\mathbf{q}^T \log \boldsymbol{\ell}$ term *exactly* when $\mathbf{q}$ is an unfactorized mixture distribution, its gradient can be computed exactly. The unbiased gradient estimator of $\mathbf{q}^T \log \mathbf{q}$ is expressed as follows[2]

$$\frac{\partial}{\partial \theta} \mathbf{q}^T \log \mathbf{q} = \frac{1}{2} \mathbf{q}^T \left( \frac{\partial}{\partial \theta} \big( \log \mathbf{q} + 1 \big)^2 \right) \approx \frac{\partial}{\partial \theta} \frac{1}{2t} \sum_{i=1}^{t} \big( \log q(\mathbf{s}_i) + 1 \big)^2, \qquad (11)$$

where $\mathbf{s}_i \in \mathbb{R}^b$ is the $i$th of $t$ samples from the variational distribution used in the Monte Carlo gradient estimator. It is evident that this surrogate loss can be easily optimized using automatic differentiation, and the per-sample computations are extremely cheap.

## 5 Numerical Studies

### 5.1 Comparison with REINFORCE

As discussed in section 2, we cannot reparameterize because of the discrete latent variable priors considered, however, we can directly compare the optimization performance of the proposed techniques with the REINFORCE gradient estimator [11]. In fig. 1, we compare ELBO maximization performance between the proposed DIRECT, and the REINFORCE methods. For this study we generated a dataset from a random weighting of $b = 20$ random Fourier features of a squared exponential kernel [31] and corrupted by independent Gaussian noise. We use a generalized linear regression model as described in section 3.1 which uses the same features with $\bar{m} = 3$. We consider a prior over $\sigma^2$, and a mean-field variational distribution giving $\bar{m}(b + 1) = 63$ variational parameters which we initialize to be the same as the prior; a uniform categorical distribution. For DIRECT, a L-BFGS optimizer is used [32] and stochastic gradient descent is used for REINFORCE with a varying number of samples used for the Monte Carlo gradient estimator. Both methods use full batch training and are implemented using TensorFlow [33]. It can be seen that DIRECT greatly outperforms REINFORCE both in the number of iterations and computational time. As we move to a large $n$ or a larger $b$, the difference between the proposed DIRECT technique and REINFORCE becomes more profound. The superior scaling with respect to $n$ was expected since we had shown in section 3.1 that the DIRECT computational runtime is independent of $n$. However, the improved scaling with respect to $b$ is an interesting result and may be attributed to the fact that as the dimension of the variational parameter space increases, there is more value in having low (or zero) variance estimates of the gradient.

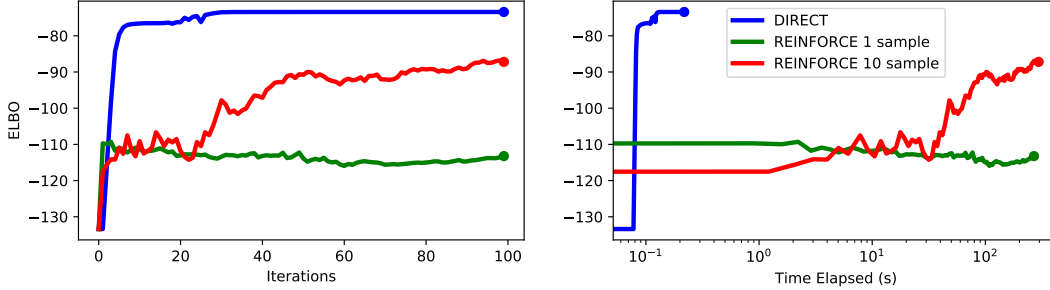

Figure 1: Convergence rates of a GLM trained with REINFORCE verses the proposed DIRECT method. The DIRECT method greatly outperforms REINFORCE in iterations and wall-clock time.

## 5.2 Relaxing Gaussian Priors on UCI Regression Datasets

In this section, we consider discretely relaxing a continuous Gaussian prior on the weights of a generalized linear regression model. This allows us to compare performance between a reparameterization gradient estimator for a continuous prior and our DIRECT method for a relaxed, discrete prior.

Considering regression datasets from the UCI repository, we report the mean and standard deviation of the root mean squared error (RMSE) from 10-fold cross validation[3]. Also presented is the mean training time per fold on a machine with two E5-2680 v3 processors and 128Gb of RAM, and the expected sparsity (percentage of zeros) within a posterior sample. All models use $b = 2000$ basis functions. Further details of the experimental setup can be found in appendix E. In table 1, we see the results of our studies across several model-types. In the left column, the "REPARAM Mean-Field" model uses a (continuous) Gaussian prior, an uncorrelated Gaussian variational distribution and reparameterization gradients. The right two models use a discrete relaxation of a Gaussian prior (DIRECT) with support at 15 discrete values, allowing storage of each latent variable sample as a vector of 4-bit quantized integers. Therefore, each ELBO evaluation requires $15^{2000} > 10^{2352}$ log-likelihood evaluations, however, these computation can be done quickly by exploiting Kronecker matrix algebra. We compute the ELBO as described in section 3.1 for the "DIRECT Mean-Field" model, and use the low-variance, unbiased gradient estimator described in eq. (11) for the "DIRECT 5-Mixture SGD" model which uses a mixture distribution with $r = 5$ components, and $t = 3000$ Monte Carlo samples for the entropy gradient estimator.

The boldface entries indicate top performance on each dataset, where it is evident that the DIRECT method not only outperformed REPARAM on most datasets but also trained much faster, particularly on the large datasets due to the independence of dataset size on computational complexity. The DIRECT mean-field model contains $\bar{m}b = 30,000$ variational parameters, however, training took just seconds on all datasets, including *electric* with over 2 million points. The DIRECT mixture model contains $\bar{m}br = 150,000$ variational parameters, and since the gradient estimates are stochastic, average training times are on the order of hundreds of seconds across all datasets. While the time for precomputations does depend on dataset size, its contribution to the overall timings are negligible, being well under one second for the largest dataset, *electric*. Additionally, it is evident that posterior samples from the DIRECT model tend to be very sparse. For example, the DIRECT models on the *gas* dataset admit posterior samples that are over 84% sparse on average, meaning that over 1680 weights are expected to be zero in a posterior sample with $b = 2000$ elements. This would yield massive computational savings on hardware limited devices. Samples from the DIRECT models on the *electric* dataset are over 99.6% sparse.

Comparing the DIRECT mean-field model to the mixture model, we observe gains in the RMSE performance on many datasets, as we would expect with the increased flexibility of the variational distribution. While we only showed the posterior mean in our results, we would expect an even larger disparity in the quality of the predictive uncertainty which was not analyzed. In table 2 of the supplement, we present results for a DIRECT mixture model that uses the ELBO lower bound presented in Theorem 3. This model does not perform as well as the DIRECT mixture model trained using an unbiased SGD approach, as would be expected, however, it does train faster since its

| | | | Continuous Prior | | | Discrete 4-bit Prior | | | | |
| | | | REPARAM Mean-Field | | | DIRECT Mean-Field | | | DIRECT 5-Mixture SGD | |
| Dataset | $n$ | $d$ | Time | RMSE | Sparsity | Time | RMSE | Sparsity | RMSE | Sparsity |
|---|---|---|---|---|---|---|---|---|---|---|
| challenger | 23 | 4 | 8 | **0.515 ± 0.284** | 0% | 1 | 0.523 ± 0.248 | 17% | 0.525 ± 0.246 | 17% |
| fertility | 100 | 9 | 8 | 0.161 ± 0.043 | 0% | 2 | **0.159 ± 0.041** | 17% | 0.16 ± 0.041 | 17% |
| automobile | 159 | 25 | 5 | 0.425 ± 0.2 | 0% | 10 | 0.129 ± 0.063 | 51% | **0.122 ± 0.056** | 51% |
| servo | 167 | 4 | 5 | 0.524 ± 0.184 | 0% | 10 | **0.271 ± 0.08** | 35% | 0.274 ± 0.077 | 35% |
| cancer | 194 | 33 | 5 | 27.488 ± 5.45 | 0% | 4 | 22.954 ± 3.09 | 19% | **22.937 ± 3.135** | 19% |
| hardware | 209 | 7 | 5 | 1.796 ± 1.537 | 0% | 11 | **0.401 ± 0.048** | 51% | **0.401 ± 0.046** | 51% |
| yacht | 308 | 6 | 5 | 0.815 ± 0.17 | 0% | 1 | 0.234 ± 0.07 | 96% | **0.225 ± 0.082** | 96% |
| autompg | 392 | 7 | 5 | 4.05 ± 0.739 | 0% | 10 | 2.564 ± 0.363 | 31% | **2.543 ± 0.362** | 31% |
| housing | 506 | 13 | 5 | 3.014 ± 0.567 | 0% | 10 | **2.752 ± 0.405** | 40% | **2.699 ± 0.361** | 39% |
| forest | 517 | 12 | 5 | 1.378 ± 0.148 | 0% | 2 | 1.363 ± 0.15 | 17% | **1.357 ± 0.155** | 17% |
| stock | 536 | 11 | 5 | 0.751 ± 0.338 | 0% | 8 | 0.011 ± 0.003 | 98% | **0.008 ± 0.001** | 98% |
| pendulum | 630 | 9 | 5 | 1.465 ± 0.26 | 0% | 1 | 1.329 ± 0.282 | 68% | **1.312 ± 0.253** | 63% |
| energy | 768 | 8 | 5 | 78.852 ± 21.73 | 0% | 1 | 3.272 ± 0.332 | 99% | **2.911 ± 0.309** | 99% |
| concrete | 1030 | 8 | 5 | 10.347 ± 2.847 | 0% | 10 | **5.316 ± 0.716** | 82% | 5.477 ± 0.632 | 82% |
| solar | 1066 | 10 | 5 | 0.902 ± 0.171 | 0% | 10 | **0.787 ± 0.192** | 23% | 0.788 ± 0.189 | 23% |
| airfoil | 1503 | 5 | 5 | **2.071 ± 0.271** | 0% | 11 | 2.175 ± 0.349 | 48% | 2.156 ± 0.316 | 45% |
| wine | 1599 | 11 | 5 | 0.939 ± 0.33 | 0% | 11 | 0.472 ± 0.044 | 54% | **0.469 ± 0.042** | 54% |
| gas | 2565 | 128 | 5 | 0.27 ± 0.052 | 0% | 1 | 0.211 ± 0.058 | 84% | **0.184 ± 0.063** | 76% |
| skillcraft | 3338 | 19 | 46 | 0.273 ± 0.029 | 0% | 7 | **0.253 ± 0.016** | 97% | **0.253 ± 0.016** | 97% |
| sml | 4137 | 26 | 47 | **0.327 ± 0.013** | 0% | 1 | 0.677 ± 0.044 | 57% | 0.671 ± 0.047 | 57% |
| parkinsons | 5875 | 20 | 48 | **0.158 ± 0.009** | 0% | 1 | 0.651 ± 0.034 | 13% | 0.613 ± 0.083 | 13% |
| poletele | 15000 | 26 | 50 | **12.487 ± 0.363** | 0% | 10 | 13.65 ± 0.348 | 16% | 13.369 ± 0.431 | 17% |
| elevators | 16599 | 18 | 51 | 0.247 ± 0.156 | 0% | 1 | **0.124 ± 0.003** | 99% | **0.124 ± 0.003** | 99% |
| protein | 45730 | 9 | 58 | 0.642 ± 0.006 | 0% | 11 | 0.619 ± 0.007 | 76% | **0.618 ± 0.007** | 60% |
| kegg | 48827 | 20 | 58 | **0.178 ± 0.012** | 0% | 1 | 0.222 ± 0.009 | 96% | 0.205 ± 0.004 | 95% |
| ctslice | 53500 | 385 | 61 | **4.415 ± 0.113** | 0% | 2 | 6.063 ± 0.122 | 19% | 5.478 ± 0.137 | 42% |
| keggu | 63608 | 27 | 61 | **0.122 ± 0.004** | 0% | 1 | 0.139 ± 0.004 | 87% | 0.136 ± 0.006 | 87% |
| 3droad | 434874 | 3 | 141 | 11.057 ± 0.091 | 0% | 2 | 10.493 ± 0.105 | 40% | **10.354 ± 0.077** | 33% |
| song | 515345 | 90 | 158 | 0.537 ± 0.002 | 0% | 2 | 0.501 ± 0.002 | 32% | **0.498 ± 0.002** | 28% |
| buzz | 583250 | 77 | 169 | **0.94 ± 0.006** | 0% | 1 | 1.007 ± 0.007 | 82% | 0.959 ± 0.004 | 80% |
| electric | 2049280 | 11 | 500 | 9.26 ± 4.47 | 0% | 1 | 0.575 ± 0.032 | 99.6% | **0.557 ± 0.055** | 99.6% |

Table 1: Mean and standard deviation of test error, average training time, and average expected sparsity of a posterior sample from 10-fold cross validation on UCI regression datasets.

objective is evaluated deterministically, and its RMSE performance is still marginally better than the DIRECT mean-field model on many datasets.

# 6 Conclusions

We have shown that by discretely relaxing continuous priors, variational inference can be performed accurately and efficiently using our DIRECT method. We have demonstrated that through the use of Kronecker matrix algebra, the ELBO of a discretely relaxed model can be efficiently and exactly computed even when this computation requires significantly more log-likelihood evaluations than the number of atoms in the known universe. Through this ability to exactly perform ELBO computations we achieve unbiased, zero-variance gradient estimates using automatic differentiation which we show significantly outperforms competing Monte Carlo alternatives that admit high-variance gradient estimates. We also demonstrate that the computational complexity of ELBO computations is *independent* of the quantity of training data using the DIRECT method, making the proposed approaches amenable to big data applications. At inference time, we show that we can again use Kronecker matrix algebra to exactly compute the statistical moments of the parameterized predictive posterior distribution, unlike competing techniques which rely on Monte Carlo sampling. Finally, we discuss and demonstrate how posterior samples can be sparse and can be represented as quantized integer values to enable efficient inference which is particularly powerful on hardware limited devices, or if energy efficiency is a major concern.

We illustrate the DIRECT approach on several popular models such as mean-field variational inference for generalized linear models and deep Bayesian neural networks for regression. We also discuss some models which do not admit a compact representation for exact ELBO computations. For these cases, we discuss and demonstrate novel extensions to the DIRECT method that allow efficient computation of a lower bound of the ELBO, and we demonstrate how an unfactorized variational distribution can be used by introducing a manageable level of stochasticity into the gradients. We hope that these new approaches for ELBO computations will inspire new model structures and research directions in approximate Bayesian inference.

## Acknowledgements

Research funded by an NSERC Discovery Grant and the Canada Research Chairs program.

## Footnotes

[1] The discrete values that the $i$th latent variable can take, $\bar{\mathbf{w}}_i$, may be chosen a priori or learned during ELBO maximization (may be helpful for coarse discretizations). For the sake of simplicity, we focus on the former.

[2]We used the identity $\big( \log \mathbf{q} + 1 \big) \odot \frac{\partial \log \mathbf{q}}{\partial \theta} = \frac{1}{2} \frac{\partial}{\partial \theta} \big( \log \mathbf{q} + 1 \big)^2$, where $\odot$ denotes an elementwise product.

[3]90% train, 10% test per fold. We use folds from `https://people.orie.cornell.edu/andrew/code`

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
