[Supplementary Material]

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

 | 23 | 4 | 8 | $\mathbf{0.515 \pm 0.284}$ | 0% | 1 | $0.523 \pm 0.248$ | 17% | $0.525 \pm 0.246$ | 17% |
| fertility | 100 | 9 | 8 | $0.161 \pm 0.043$ | 0% | 2 | $\mathbf{0.159 \pm 0.041}$ | 17% | $0.16 \pm 0.041$ | 17% |
| automobile | 159 | 25 | 5 | $0.425 \pm 0.2$ | 0% | 10 | $0.129 \pm 0.063$ | 51% | $\mathbf{0.122 \pm 0.056}$ | 51% |
| servo | 167 | 4 | 5 | $0.524 \pm 0.184$ | 0% | 10 | $\mathbf{0.271 \pm 0.08}$ | 35% | $0.274 \pm 0.077$ | 35% |
| cancer | 194 | 33 | 5 | $27.488 \pm 5.45$ | 0% | 4 | $22.954 \pm 3.09$ | 19% | $\mathbf{22.937 \pm 3.135}$ | 19% |
| hardware | 209 | 7 | 5 | $1.796 \pm 1.537$ | 0% | 11 | $\mathbf{0.401 \pm 0.048}$ | 51% | $\mathbf{0.401 \pm 0.046}$ | 51% |
| yacht | 308 | 6 | 5 | $0.815 \pm 0.17$ | 0% | 1 | $0.234 \pm 0.07$ | 96% | $\mathbf{0.225 \pm 0.082}$ | 96% |
| autompg | 392 | 7 | 5 | $4.05 \pm 0.739$ | 0% | 10 | $2.564 \pm 0.363$ | 31% | $\mathbf{2.543 \pm 0.362}$ | 31% |
| housing | 506 | 13 | 5 | $3.014 \pm 0.567$ | 0% | 10 | $\mathbf{2.752 \pm 0.405}$ | 40% | $2.699 \pm 0.361$ | 39% |
| forest | 517 | 12 | 5 | $1.378 \pm 0.148$ | 0% | 2 | $1.363 \pm 0.15$ | 17% | $\mathbf{1.357 \pm 0.155}$ | 17% |
| stock | 536 | 11 | 5 | $0.751 \pm 0.338$ | 0% | 8 | $0.011 \pm 0.003$ | 98% | $\mathbf{0.008 \pm 0.001}$ | 98% |
| pendulum | 630 | 9 | 5 | $1.465 \pm 0.26$ | 0% | 1 | $1.329 \pm 0.282$ | 68% | $\mathbf{1.312 \pm 0.253}$ | 63% |
| energy | 768 | 8 | 5 | $78.852 \pm 21.73$ | 0% | 1 | $3.272 \pm 0.332$ | 99% | $\mathbf{2.911 \pm 0.309}$ | 99% |
| concrete | 1030 | 8 | 5 | $10.347 \pm 2.847$ | 0% | 10 | $\mathbf{5.316 \pm 0.716}$ | 82% | $5.477 \pm 0.632$ | 82% |
| solar | 1066 | 10 | 5 | $0.902 \pm 0.171$ | 0% | 10 | $\mathbf{0.787 \pm 0.192}$ | 23% | $0.788 \pm 0.189$ | 23% |
| airfoil | 1503 | 5 | 5 | $\mathbf{2.071 \pm 0.271}$ | 0% | 11 | $2.175 \pm 0.349$ | 48% | $2.156 \pm 0.316$ | 45% |
| wine | 1599 | 11 | 5 | $0.939 \pm 0.33$ | 0% | 11 | $0.472 \pm 0.044$ | 54% | $\mathbf{0.469 \pm 0.042}$ | 54% |
| gas | 2565 | 128 | 5 | $0.27 \pm 0.052$ | 0% | 1 | $0.211 \pm 0.058$ | 84% | $\mathbf{0.184 \pm 0.063}$ | 76% |
| skillcraft | 3338 | 19 | 46 | $0.273 \pm 0.029$ | 0% | 7 | $\mathbf{0.253 \pm 0.016}$ | 97% | $\mathbf{0.253 \pm 0.016}$ | 97% |
| sml | 4137 | 26 | 47 | $\mathbf{0.327 \pm 0.013}$ | 0% | 1 | $0.677 \pm 0.044$ | 57% | $0.671 \pm 0.047$ | 57% |
| parkinsons | 5875 | 20 | 48 | $\mathbf{0.158 \pm 0.009}$ | 0% | 1 | $0.651 \pm 0.034$ | 13% | $0.613 \pm 0.083$ | 13% |
| poletele | 15000 | 26 | 50 | $\mathbf{12.487 \pm 0.363}$ | 0% | 10 | $13.65 \pm 0.348$ | 16% | $13.369 \pm 0.431$ | 17% |
| elevators | 16599 | 18 | 51 | $0.247 \pm 0.156$ | 0% | 1 | $\mathbf{0.124 \pm 0.003}$ | 99% | $\mathbf{0.124 \pm 0.003}$ | 99% |
| protein | 45730 | 9 | 58 | $0.642 \pm 0.006$ | 0% | 11 | $0.619 \pm 0.007$ | 76% | $\mathbf{0.618 \pm 0.007}$ | 60% |
| kegg | 48827 | 20 | 58 | $\mathbf{0.178 \pm 0.012}$ | 0% | 1 | $0.222 \pm 0.009$ | 96% | $0.205 \pm 0.004$ | 95% |
| ctslice | 53500 | 385 | 61 | $\mathbf{4.415 \pm 0.113}$ | 0% | 2 | $6.063 \pm 0.122$ | 19% | $5.478 \pm 0.137$ | 42% |
| keggu | 63608 | 27 | 61 | $\mathbf{0.122 \pm 0.004}$ | 0% | 1 | $0.139 \pm 0.004$ | 87% | $0.136 \pm 0.006$ | 87% |
| 3droad | 434874 | 3 | 141 | $11.057 \pm 0.091$ | 0% | 2 | $10.493 \pm 0.105$ | 40% | $\mathbf{10.354 \pm 0.077}$ | 33% |
| song | 515345 | 90 | 158 | $0.537 \pm 0.002$ | 0% | 2 | $0.501 \pm 0.002$ | 32% | $\mathbf{0.498 \pm 0.002}$ | 28% |
| buzz | 583250 | 77 | 169 | $\mathbf{0.94 \pm 0.006}$ | 0% | 1 | $1.007 \pm 0.007$ | 82% | $0.959 \pm 0.004$ | 80% |
| electric | 2049280 | 11 | 500 | $9.26 \pm 4.47$ | 0% | 1 | $0.575 \pm 0.032$ | 99.6% | $\mathbf{0.557 \pm 0.055}$ | 99.6% |

Table 1: Mean and standard deviation of test error, average training time, and average expected sparsity of a posterior sample from 10-fold cross validation on UCI regression datasets.

objective is evaluated deterministically, and its RMSE performance is still marginally better than the DIRECT mean-field model on many datasets.

## 6   Conclusions

We have shown that by discretely relaxing continuous priors, variational inference can be performed accurately and efficiently using our DIRECT method. We have demonstrated that through the use of Kronecker matrix algebra, the ELBO of a discretely relaxed model can be efficiently and exactly computed even when this computation requires significantly more log-likelihood evaluations than the number of atoms in the known universe. Through this ability to exactly perform ELBO computations we achieve unbiased, zero-variance gradient estimates using automatic differentiation which we show significantly outperforms competing Monte Carlo alternatives that admit high-variance gradient estimates. We also demonstrate that the computational complexity of ELBO computations is *independent* of the quantity of training data using the DIRECT method, making the proposed approaches amenable to big data applications. At inference time, we show that we can again use Kronecker matrix algebra to exactly compute the statistical moments of the parameterized predictive posterior distribution, unlike competing techniques which rely on Monte Carlo sampling. Finally, we discuss and demonstrate how posterior samples can be sparse and can be represented as quantized integer values to enable efficient inference which is particularly powerful on hardware limited devices, or if energy efficiency is a major concern.

We illustrate the DIRECT approach on several popular models such as mean-field variational inference for generalized linear models and deep Bayesian neural networks for regression. We also discuss some models which do not admit a compact representation for exact ELBO computations. For these cases, we discuss and demonstrate novel extensions to the DIRECT method that allow efficient computation of a lower bound of the ELBO, and we demonstrate how an unfactorized variational distribution can be used by introducing a manageable level of stochasticity into the gradients. We hope that these new approaches for ELBO computations will inspire new model structures and research directions in approximate Bayesian inference.

## Acknowledgements

Research funded by an NSERC Discovery Grant and the Canada Research Chairs program.

## Footnotes

[1] The discrete values that the $i$th latent variable can take, $\bar{\mathbf{w}}_i$, may be chosen a priori or learned during ELBO maximization (may be helpful for coarse discretizations). For the sake of simplicity, we focus on the former.

[2]We used the identity $\left( \log \mathbf{q} + 1 \right) \odot \frac{\partial \log \mathbf{q}}{\partial \theta} = \frac{1}{2} \frac{\partial}{\partial \theta} \left( \log \mathbf{q} + 1 \right)^2$, where $\odot$ denotes an elementwise product.

[3]90% train, 10% test per fold. We use folds from `https://people.orie.cornell.edu/andrew/code`

[4]90% train, 10% test per fold. We use folds from `https://people.orie.cornell.edu/andrew/code`

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

# A   Proof of Theorem 1: ELBO Computation for a Regression GLM

For our generalized linear regression model with a prior over $\sigma^2$, we can re-write eq. (7) as follows

$$\mathsf{ELBO}(\boldsymbol{\theta}) = (\mathbf{q}_\sigma \otimes \mathbf{q})^T \log \boldsymbol{\ell} + \sum_{i=1}^{b} \mathbf{q}_i^T \log \mathbf{p}_i - \sum_{i=1}^{b} \mathbf{q}_i^T \log \mathbf{q}_i + \mathbf{q}_\sigma^T \log \mathbf{p}_\sigma - \mathbf{q}_\sigma^T \log \mathbf{q}_\sigma, \quad (12)$$

where we have simply expanded the factorized variational distribution to include $\sigma^2$, resulting in the two extra terms. To complete the ELBO in eq. (12), we need to take the inner product between the variational distribution and log-likelihood for each point in the hypothesis space, $(\mathbf{q}_\sigma \otimes \mathbf{q})^T \log \boldsymbol{\ell}$. We can write this relation as follows for our generalized linear regression model, (see e.g. [34])

$$(\mathbf{q}_\sigma \otimes \mathbf{q})^T \log \boldsymbol{\ell} = -\frac{n}{2} \mathbf{q}_\sigma^T \log \boldsymbol{\sigma}^2 - \frac{1}{2} \big(\mathbf{q}_\sigma^T \boldsymbol{\sigma}^{-2}\big) \big(\mathbf{q}^T \{(\mathbf{y} - \boldsymbol{\Phi}\mathbf{w}_i)^T (\mathbf{y} - \boldsymbol{\Phi}\mathbf{w}_i)\}_{i=1}^m \big), \quad (13)$$

whose computation would be prohibitively expensive when $m = \bar{m}^b$ is large. We will now focus on computing the inner product involving the variational distribution over the $\mathbf{w}$ variables, $\mathbf{q}$, which we can break into three terms as follows,

$$\mathbf{q}^T \{(\mathbf{y} - \boldsymbol{\Phi}\mathbf{w}_i)^T (\mathbf{y} - \boldsymbol{\Phi}\mathbf{w}_i)\}_{i=1}^m = \mathbf{y}^T \mathbf{y} - 2\mathbf{q}^T \{\mathbf{y}^T \boldsymbol{\Phi}\mathbf{w}_i\}_{i=1}^m + \mathbf{q}^T \{\mathbf{w}_i^T \boldsymbol{\Phi}^T \boldsymbol{\Phi}\mathbf{w}_i\}_{i=1}^m, \quad (14)$$

for which the first term is trivial to compute as written since it does not depend on $\mathbf{w}$. We now demonstrate how the second and third terms can be computed, recalling that we have assumed that $\mathbf{q}$ is a mean-field variational distribution. Firstly, define $\mathbf{Z} = (\boldsymbol{\Phi}\mathbf{W})^T = \{\oplus_{j=1}^b \phi_{ij}\bar{\mathbf{w}}_j\}_{i=1}^n \in \mathbb{R}^{m \times n}$ whose columns contain the model prediction for a single training point at every possible set of latent variable values in the hypothesis space. Observe that each column is represented as a sum of $b$ Kronecker product vectors. We can then write the second term of eq. (14) as

$$\mathbf{q}^T \{\mathbf{y}^T \boldsymbol{\Phi}\mathbf{w}_i\}_{i=1}^m = \sum_{i=1}^n y_i \mathbf{q}^T \mathbf{z}_i = \sum_{i=1}^n y_i \sum_{j=1}^b \phi_{ij} \mathbf{q}_j^T \bar{\mathbf{w}}_j = \sum_{j=1}^b \mathbf{q}_j^T \bar{\mathbf{w}}_j \bigg(\sum_{i=1}^n y_i \phi_{ij}\bigg) = \mathbf{s}^T \big(\boldsymbol{\Phi}^T \mathbf{y}\big), \quad (15)$$

where $\mathbf{s} = \{\mathbf{q}_j^T \bar{\mathbf{w}}_j\}_{j=1}^b \in \mathbb{R}^b$. Finally, considering the third term of eq. (14), observe that we can write $\{\mathbf{w}_i^T \boldsymbol{\Phi}^T \boldsymbol{\Phi}\mathbf{w}_i\}_{i=1}^m = \sum_{i=1}^n \mathbf{z}_i^2$, and since each $\mathbf{z}_i$ is a sum of $b$ Kronecker product vectors, $\mathbf{z}_i^2$ a sum of $(b + b^2)/2$ Kronecker product vectors. We can then write the third term of eq. (14) as follows,

$$\mathbf{q}^T \{\mathbf{w}_i^T \boldsymbol{\Phi}^T \boldsymbol{\Phi}\mathbf{w}_i\}_{i=1}^m = \sum_{i=1}^n \sum_{j=1}^b \mathbf{q}_j^T \bar{\mathbf{w}}_j^2 \phi_{ij}^2 + 2 \sum_{k=j+1}^b \phi_{ij}\phi_{ik}(\mathbf{q}_j^T \bar{\mathbf{w}}_j)(\mathbf{q}_k^T \bar{\mathbf{w}}_k), \quad (16)$$

$$= \sum_{j=1}^b \mathbf{q}_j^T \bigg(\bar{\mathbf{w}}_j^2 \sum_{i=1}^n \phi_{ij}^2\bigg) + 2 \sum_{k=j+1}^b s_j s_k \bigg(\sum_{i=1}^n \phi_{ij}\phi_{ik}\bigg), \quad (17)$$

$$= \mathbf{s}^T \boldsymbol{\Phi}^T \boldsymbol{\Phi}\mathbf{s} - \mathtt{diag}(\boldsymbol{\Phi}^T \boldsymbol{\Phi})^T \mathbf{s}^2 + \sum_{j=1}^b \mathbf{q}_j^T \mathbf{h}_j, \quad (18)$$

where we have used the short-hand notation $\mathbf{H} = \{\bar{\mathbf{w}}_j^2 \sum_{i=1}^n \phi_{ij}^2\}_{j=1}^b \in \mathbb{R}^{\bar{m} \times b}$. Substituting eq. (15) and eq. (18) into eq. (14), we can re-write the inner product between the variational distribution and the log-likelihood in eq. (13) as follows,

$$(\mathbf{q}_\sigma \otimes \mathbf{q})^T \log \boldsymbol{\ell} = -\frac{n}{2} \mathbf{q}_\sigma^T \log \boldsymbol{\sigma}^2 - \frac{1}{2} \big(\mathbf{q}_\sigma^T \boldsymbol{\sigma}^{-2}\big) \Big(\mathbf{y}^T \mathbf{y} - 2\mathbf{s}^T \big(\boldsymbol{\Phi}^T \mathbf{y}\big) + \mathbf{s}^T \boldsymbol{\Phi}^T \boldsymbol{\Phi}\mathbf{s} -$$

$$\mathtt{diag}(\boldsymbol{\Phi}^T \boldsymbol{\Phi})^T \mathbf{s}^2 + \sum_{j=1}^b \mathbf{q}_j^T \mathbf{h}_j\Big), \quad (19)$$

and substituting this into eq. (12) completes the proof. □

# B   Proof of Theorem 2: Logistic Regression ELBO Lower Bound

For the generalized linear logistic regression model considered, we can write the log likelihood as follows (see e.g. [34])

$$\log \boldsymbol{\ell} = \sum_{i=1}^n -y_i \mathbf{z}_i - \log \big(1 + \exp(-\mathbf{z}_i)\big), \quad (20)$$

where $\mathbf{Z} = (\mathbf{\Phi}\mathbf{W})^T = \{\oplus_{j=1}^b \phi_{ij}\bar{\mathbf{w}}_j\}_{i=1}^n \in \mathbb{R}^{m \times n}$ is a matrix whose columns contain the logit values for a single training point at every possible set of latent variables in the hypothesis space. It is evident that the first term is identical to that discussed in eq. (15), however, computation of the second term requires more development. We can write

$$\mathbf{q}^T \log \boldsymbol{\ell} = -\mathbf{s}^T(\mathbf{\Phi}^T\mathbf{y}) - \sum_{i=1}^n \mathbf{q}^T \log\left(1 + \exp(-\mathbf{z}_i)\right). \tag{21}$$

Since $\mathbf{z}_i = \oplus_{j=1}^b \phi_{ij}\bar{\mathbf{w}}_j \in \mathbb{R}^m$ is a sum of $b$ Kronecker product vectors, each with one unique sub-matrix that is not unity, $\exp(-\mathbf{z}_i)$ is a single Kronecker product vector. This follows from Proposition 2. We can then take a Taylor series explanation of $\log\left(1 + \exp(-\mathbf{z}_i)\right)$ as follows

$$\log(1 + \exp(-\mathbf{z}_i)) = -\sum_{k=1}^\infty \frac{(-1)^k \exp(-k\mathbf{z}_i)}{k} \qquad \text{for } |\exp(-\mathbf{z}_i)| < 1 \to \mathbf{z}_i > 0, \tag{22}$$

$$\log(1 + \exp(-\mathbf{z}_i)) = -\mathbf{z}_i - \sum_{k=1}^\infty \frac{(-1)^k \exp(k\mathbf{z}_i)}{k} \qquad \text{for } |\exp(-\mathbf{z}_i)| > 1 \to \mathbf{z}_i < 0, \tag{23}$$

and although the use of either choice would result in an ELBO lower bound, we choose the approximation based on the training label as follows; if $y_i = 0$ or 1 then we would choose the $(\mathbf{z}_i > 0)$ or $(\mathbf{z}_i < 0)$ approximation, respectively. We choose this because $z_i > 0$ gives a higher class conditional probability to class 0 than class 1 so this approximation would yield a tight lower bound when the training examples are correctly classified. These approximations are plotted in fig. 2 with a first-order expansion where it is evident that the computation lower-bounds the exact computation. Using this first-order Taylor series approximation, we can write our lower bound for the inner product between the variational distribution and the log-likelihood as follows which completes the proof,

$$\mathbf{q}^T \log \boldsymbol{\ell} \geqslant -\mathbf{s}^T(\mathbf{\Phi}^T\mathbf{y}) - \sum_{i=1}^n \begin{cases} \mathbf{q}^T \exp(-\mathbf{z}_i) & \text{if } y_i = 0 \\ \mathbf{q}^T \exp(\mathbf{z}_i) - \mathbf{q}^T\mathbf{z}_i & \text{if } y_i = 1 \end{cases}, \tag{24}$$

$$= -\mathbf{s}^T(\mathbf{\Phi}^T\mathbf{y}) - \sum_{i=1}^n \begin{cases} \prod_{j=1}^b \mathbf{q}_j^T \exp(-\phi_{ij}\bar{\mathbf{w}}_j) & \text{if } y_i = 0 \\ \prod_{j=1}^b \mathbf{q}_j^T \exp(\phi_{ij}\bar{\mathbf{w}}_j) - \sum_{j=1}^b \mathbf{q}_i^T \phi_{ij}\bar{\mathbf{w}}_j & \text{if } y_i = 1 \end{cases}. \tag{25}$$

$\square$

**Remark**   We expect these Taylor series approximations to admit a tight bound within and just outside of their logit domains as we can see in fig. 2. Equivalently, the log-likelihood approximation is accurately computed for hypotheses that correctly classify the training data when we use this lower bound, however, hypotheses that confidently misclassify training labels may be over-penalized. This can be seen by observing how the approximations in fig. 2 significantly underestimate the exact solution when they are far outside of the approximations domain.

**Remark**   In practice, the products over $b$ terms in Theorem 2 may result in overflow or loss of precision, however, computations can be performed in a stable manner in logit space and the LogSumExp trick [35] can be used to avoid precision loss for the sum over $n$.

## C   Proof of Theorem 3: Mixture Distribution Entropy Lower Bound

We begin by taking a Taylor series approximation of $\log \mathbf{q}$ about $\mathbf{a} = \otimes_{i=1}^b \mathbf{a}_i$, $\mathbf{a}_i \in (0,1)^{\overline{m}}$ as follows,

$$\log \mathbf{q} = \log \mathbf{a} + \sum_{k=1}^\infty \frac{(-1)^{(k+1)}}{k\mathbf{a}^k}(\mathbf{q} - \mathbf{a})^k, \tag{26}$$

which can be represented as a sum of Kronecker product vectors once the exponents are computed explicitly. However, the number of terms in this sum will grow quickly with respect to the order of the Taylor series approximation. When a first order Taylor series expansion is considered, the approximation will give a strict lower bound of $-\log \mathbf{q}$ and consequently a lower bound of the ELBO (eq. (7)) will be achieved. The approximation for a linear Taylor series expansion is plotted in

Figure 2: First-order Taylor series approximation of $-\log\big(1 + \exp(-z)\big)$. The approximations evidently lower-bound the exact computation.

Figure 3: Taylor series approximation of $-\log(q)$ about $a = 0.5$. The approximations evidently lower-bound the exact computation.

fig. 3 where it is apparent that the approximation lower-bounds the exact computation. We consider this linear approximation for the result in Theorem 3. Note that the exact computation will always be lower bounded irrespective of the location that the Taylor series is taken about, therefore, we may select the values of $\{\mathbf{a}_i \in (0,1)^{\overline{m}}\}_{i=1}^b$ that maximize this lower bound, as written in the theorem statement. We can then write our approximation of the third term from the ELBO (eq. (7)) to complete the proof as follows

$$-\mathbf{q}^T \log \mathbf{q} \geqslant 1 - \sum_{j=1}^{r} \alpha_j \Bigg( \sum_{i=1}^{b} \mathbf{q}_i^{(j)\,T} \log \mathbf{a}_i + \alpha_j \prod_{i=1}^{b} \mathbf{q}_i^{(j)\,T} \frac{\mathbf{q}_i^{(j)}}{\mathbf{a}_i} + 2 \sum_{k=j+1}^{r} \alpha_k \prod_{i=1}^{b} \mathbf{q}_i^{(j)\,T} \frac{\mathbf{q}_i^{(k)}}{\mathbf{a}_i} \Bigg). \quad (27)$$

$\square$

**Remark** The products over $b$ terms might seem problematic, however, we do not expect the final results to be too large to be an overflow concern. To avoid precision loss, we compute the log of the products, which can be done stably, and then exponentiate.

## D   DIRECT Bayesian Neural Networks for Regression

In order to demonstrate DIRECT computation of the log-likelihood for a Bayesian neural network we will first perform a forward-pass through the neural network from top to bottom, however, unlike how a forward-pass is conventionally conducted in literature where the network is fixed at a specific location in the hypothesis space, we will simultaneously evaluate the neural network at *all* locations in entire hypothesis space. Consequently, a forward-pass through the neural network with our $n$-point training set will give us $\overline{m}^b \times n$ values.

**Nomenclature and Neuron Structure** At all points in the forward-pass we can represent the internal (or final) state of the neural network with a special structure which is a sum of Kronecker product vectors as follows for $i = 1, \dots,$ (number of neurons in the layer), and $l = 1, \dots, n$,

$$\mathbf{u}_l^{(i)} = \sum_{j=1}^{h} c_{jl} \bigotimes_{k=1}^{b} \mathbf{g}_{jk}^{(i)}, \quad (28)$$

where $\mathbf{U}^{(i)} = \{\mathbf{u}_l^{(i)}\}_{l=1}^n \in \mathbb{R}^{\overline{m}^b \times n}$, $\mathbf{u}_l^{(i)} \in \mathbb{R}^{\overline{m}^b}$ denotes the internal state of the $i$th neuron of the current layer, and both $\mathbf{G}^{(i)} = \{\{\mathbf{g}_{jk}^{(i)}\}_{j=1}^h\}_{k=1}^b \in \mathbb{R}^{h \times b \times \overline{m}}$, $\mathbf{g}_{jk}^{(i)} \in \mathbb{R}^{\overline{m}}$ and $\mathbf{C} \in \mathbb{R}^{h \times n}$ change as we move from one layer to the next. $h$ depends on the network architecture and it is constant throughout a layer but grows as we observe deeper layers. Using this nomenclature it is evident that we can compactly represent the internal state of any location within the neural network while we compute our forward pass.

The following image denotes the structure of a neuron that we will use in our neural network.

For clarity of illustration, we will not discuss the bias term although this can be easily added by associating a latent variable with a layer input that is fixed to unity. In our discussion, we will break the computation of the neuron into two stages; the first will involve multiplication of the layer inputs with the latent variables as well as the summation, and the second stage will involve passing this summation through a non-linear activation function.

**Multiplication with Latent Variables and Summation**  Our computational neurons begin by multiplying the layer inputs with a specific latent variable and then summing up these values. Assuming we are conducting a forward-pass moving deeper into the network and are currently at the "layer inputs" location in our computational neuron figure, the internal state for the $i$th input is denoted by $\mathbf{U}^{(i)} \in \mathbb{R}^{\overline{m}^b \times n}$ whose structure is defined in eq. (28). We must multiply this state by all possible values of the corresponding latent variable, which we will assume is indexed as the $p$th of our $b$ latent variables. We can easily perform this multiplication as follows for $l = 1, \ldots, n$

$$\mathbf{u}_l^{\prime(i)} = \left( \sum_{j=1}^{h} c_{jl} \bigotimes_{k=1}^{b} \mathbf{g}_{jk}^{(i)} \right) \odot \mathbf{W}[p,:]^T, \tag{29}$$

$$= \sum_{j=1}^{h} c_{jl} \left( \bigotimes_{k=1}^{p-1} \mathbf{g}_{jk}^{(i)} \right) \otimes \left( \mathbf{g}_{jp}^{(i)} \odot \bar{\mathbf{w}}_p \right) \otimes \left( \bigotimes_{k=p+1}^{b} \mathbf{g}_{jk}^{(i)} \right) = \sum_{j=1}^{h} c_{jl} \bigotimes_{k=1}^{b} \mathbf{g}_{jk}^{\prime(i)}, \tag{30}$$

where $\odot$ denotes element-wise multiplication, and we have taken advantage of the Kronecker product structure of the rows of $\mathbf{W}$ as depicted in eq. (3). Finally, the summing operation is straightforward for our computational neuron. It simply involves summing the multiplied inputs from each layer input as follows,

$$\sum_{i=1}^{\text{num. inputs}} \sum_{j=1}^{h} c_{jl} \bigotimes_{k=1}^{b} \mathbf{g}_{jk}^{\prime(i)}. \tag{31}$$

At this point we would update $h$, $\mathbf{G}$ and $\mathbf{C}$ to convert this double summation into a single summation before passing through the non-linear activation function, as we will discuss next.

**Quadratic Activation**  We will use a quadratic activation function for our neural network. Any other non-linear activation could be used, however, we choose the quadratic since it allows a more compact representation of internal state of the network to be maintained, i.e. allows for a small $h$ versus other non-linear activations. Again assuming that the current state at the $i$th neuron is defined by $\mathbf{U}^{(i)}$, the output for the activation function for the $i$th neuron is as follows for $l = 1, \ldots, n$

$$\mathbf{u}_l^{\prime(i)} = \mathbf{u}_l^{(i)} \odot \mathbf{u}_l^{(i)} = \left( \sum_{j=1}^{h} c_{jl} \bigotimes_{k=1}^{b} \mathbf{g}_{jk}^{(i)} \right) \odot \left( \sum_{j=1}^{h} c_{jl} \bigotimes_{k=1}^{b} \mathbf{g}_{jk}^{(i)} \right), \tag{32}$$

$$= \sum_{j=1}^{h} c_{jl}^2 \bigotimes_{k=1}^{b} \mathbf{g}_{jk}^{(i)} \odot \mathbf{g}_{jk}^{(i)} + 2 \sum_{j=1}^{h} \sum_{p=1}^{j-1} c_{jl} c_{pl} \mathbf{g}_{jk}^{(i)} \odot \mathbf{g}_{pk}^{(i)}, \tag{33}$$

and at this point we would update $h$, $\mathbf{G}$ and $\mathbf{C}$ to convert this double summation into a single summation to represent the internal state compactly before moving deeper.

**Forward-Pass Algorithm**   Using the previously defined operations, we can summarize the forward-pass procedure in algorithm `forward_pass`. Note that algorithm `forward_pass` is simplified for clarity of presentation. The computations involved could be performed far more efficiently and in a more stable manner. For example, the vast majority of entries in the $\mathbf{G}$ matrices are unity, so identifying this could massively decrease storage and computational requirements. Additionally, $\tilde{\mathbf{C}}$ evidently has a Kronecker product structure which could be carefully exploited to yield benefits for very wide neural networks. For stability, all matrices could be represented by storing both the sign and logarithm of all entries. For deep networks, this could be advantageous to avoid precision loss. Nonetheless, we will proceed with the algorithm as presented, for purposes of clarity.

---

**Algorithm** `forward_pass` Perform a forward pass for through the neural network using the entire training set and simultaneously computing the outputs for all $m = \bar{m}^b$ points in the hypothesis space. `mult_var` multiplies the current state with the appropriate latent variable as is done in eq. (30), `neuron_sum` computes the neuron sum as is done in eq. (31), and `activation` computes the non-linear activation function as is done in eq. (33). All the pseudo-functions defined take $\mathbf{G}$ and/or $\mathbf{C}$ and perform the necessary computations with those inputs. We omit latent-variable indexing values for clarity of presentation.

---

**Input:** $\mathbf{X} \in \mathbb{R}^{n \times d}$
**Output:** $\mathbf{C} \in \mathbb{R}^{h \times n}$ & $\mathbf{G} \in \mathbb{R}^{h \times b \times \overline{m}}$ which define state $\mathbf{U} \in \mathbb{R}^{\overline{m}^b \times n}$ in eq. (28)
$\mathbf{C} = \mathbf{X}^T, \qquad \mathbf{G}^{(i)} = \texttt{ones}(1 \times b \times \bar{m}), \; i = 1, \ldots, d$
**for** each layer **do**
$\quad \tilde{\mathbf{C}} = \texttt{neuron\_sum}(\{\mathbf{C}\}_1^{\text{num. inputs}}) = \mathbf{1}_{\text{num. inputs}} \otimes \mathbf{C}$
$\quad$ **for** $j = 1$ **to** num. neurons in layer **do**
$\quad\quad$ **for** $i = 1$ **to** num. inputs to layer **do**
$\quad\quad\quad \mathbf{G}'^{(i)} = \texttt{mult\_var}(\mathbf{G}^{(i)})$ $\qquad\qquad$ multiplication with the appropriate row of $\mathbf{W}$
$\quad\quad$ **end for**
$\quad\quad \tilde{\mathbf{G}}^{(j)} = \texttt{neuron\_sum}(\mathbf{G}'^{(1)}, \ldots, \mathbf{G}'^{(\text{num. inputs})})$ $\;$ sum operation for the current ($j$th) neuron
$\quad\quad$ **if not** last layer **then**
$\quad\quad\quad \tilde{\mathbf{G}}^{(j)}, \tilde{\mathbf{C}} = \texttt{activation}(\tilde{\mathbf{G}}^{(j)}, \tilde{\mathbf{C}})$
$\quad\quad$ **end if**
$\quad$ **end for**
$\quad \mathbf{C} = \tilde{\mathbf{C}}, \qquad \mathbf{G}^{(j)} = \tilde{\mathbf{G}}^{(j)}, j = 1, \ldots, \text{num. neurons in layer}$ $\qquad\qquad$ update variables
**end for**
$\mathbf{G} = \mathbf{G}^{(1)}$ $\qquad\qquad\qquad\qquad\qquad$ only one neuron in the last (output) layer, so remove indexing

---

**ELBO Computation**   Computation of the ELBO will proceed similarly to the GLM regression model in section 3.1, however, there are several differences since we no longer have constant basis functions so our state representation is more complicated. We will again assume a Gaussian noise model for the observed responses and will again place a prior over the Gaussian variance. We can then modify eq. (13) which focuses on the ELBO term related to the log-likelihood as follows

$$(\mathbf{q}_\sigma \otimes \mathbf{q})^T \log \boldsymbol{\ell} = -\frac{n}{2} \mathbf{q}_\sigma^T \log \boldsymbol{\sigma}^2 - \frac{1}{2}\left(\mathbf{q}_\sigma^T \boldsymbol{\sigma}^{-2}\right)\left(\mathbf{q}^T \{(\mathbf{y} - \mathbf{U}[i,:]^T)^T(\mathbf{y} - \mathbf{U}[i,:]^T)\}_{i=1}^m\right), \quad (34)$$

where we assume that we have already conducted algorithm `forward_pass` such that the state $\mathbf{U}$ represents the output of the neural network. We will now focus on computing the inner product involving the variational distribution over the $\mathbf{w}$ variables, $\mathbf{q}$, which we can break into three terms as follows,

$$\mathbf{q}^T \{(\mathbf{y} - \mathbf{U}[i,:]^T)^T(\mathbf{y} - \mathbf{U}[i,:]^T)\}_{i=1}^m =$$
$$\mathbf{y}^T \mathbf{y} - 2\mathbf{q}^T \{\mathbf{y}^T \mathbf{U}[i,:]^T\}_{i=1}^m + \mathbf{q}^T \{\mathbf{U}[i,:]\mathbf{U}[i,:]^T\}_{i=1}^m, \quad (35)$$

for which the first term is trivial to compute as written since it does not depend on the latent variables. We now demonstrate how the second and third terms can be computed, recalling we assume $\mathbf{q}$ is a mean-field variational distribution (although we can extend beyond mean-field using the techniques

discussed in section 4). Considering the second term in eq. (35), we can write

$$\mathbf{q}^T \{\mathbf{y}^T \mathbf{U}[i,:]^T\}_{i=1}^m = \mathbf{q}^T \left( \sum_{k=1}^n y_k \sum_{j=1}^h c_{jk} \bigotimes_{i=1}^b \mathbf{g}_{ij} \right) = \sum_{j=1}^h \left( \sum_{k=1}^n y_k c_{jk} \right) \prod_{i=1}^b \mathbf{q}_i^T \mathbf{g}_{ij},$$

$$= \sum_{j=1}^h p_j \prod_{i=1}^b \mathbf{q}_i^T \mathbf{g}_{ij}, \tag{36}$$

where we have used the short-hand notation $\mathbf{p} = \{\sum_{k=1}^n y_k c_{jk}\}_j \in \mathbb{R}^h$. Finally, considering the third term in eq. (35), we can write

$$\mathbf{q}^T \{\mathbf{U}[i,:] \mathbf{U}[i,:]^T\}_{i=1}^m = \mathbf{q}^T \sum_{i=1}^n \mathbf{u}_i \odot \mathbf{u}_i = \mathbf{q}^T \sum_{i=1}^n \sum_{j=1}^h \sum_{k=1}^h c_{ji} c_{ki} \bigotimes_{l=1}^b (\mathbf{g}_{jl} \odot \mathbf{g}_{kl}), \tag{37}$$

$$= \sum_{j=1}^h \sum_{k=1}^h \left( \sum_{i=1}^n c_{ji} c_{ki} \right) \prod_{l=1}^b \mathbf{q}_l^T (\mathbf{g}_{jl} \odot \mathbf{g}_{kl}), \tag{38}$$

$$= \sum_{j=1}^h \sum_{k=1}^h v_{jk} \prod_{l=1}^b \mathbf{q}_l^T (\mathbf{g}_{jl} \odot \mathbf{g}_{kl}), \tag{39}$$

where we define $\mathbf{V} = \{\sum_{i=1}^n c_{ji} c_{ki}\}_{j,k} \in \mathbb{R}^{h \times h}$. Substituting eq. (36) and eq. (39) into eq. (35), we can now compute the inner product between the variational distribution and the log-likelihood in eq. (34). The other terms required to compute the ELBO can be seen in eq. (12), and the computation of these other terms do not differ from the case of the generalized linear regression model. So we can now tractably compute the ELBO for our DIRECT Bayesian neural network.

We can pre-compute the terms $\mathbf{y}^T \mathbf{y}$, $\mathbf{p}$, and $\mathbf{V}$ before training begins (since these do not depend on the variational parameters) such that the final complexity of the DIRECT method is *independent* of the number of training points, making the proposed techniques ideal for massive datasets. Also, it is evident that each of these pre-computed terms can easily be updated as more data is observed making the techniques amenable to online learning applications. If we assume a neural network with $\ell$ hidden layers and an equal distribution of latent variables between layers, the computational complexity of the ELBO computations are $\mathcal{O}(\ell \bar{m} (b/\ell)^{4\ell})$. This can be seen by observing eq. (39) and noting that $h = \mathcal{O}((b/\ell)^{2\ell})$, and that only $\mathcal{O}(\ell)$ of the vectors in $\{\mathbf{g}_{jl}\}_{l=1}^b$ are not unity for any value of $j = 1, \ldots, h$, allowing computations to be saved.

# E    UCI Regression Studies Setup & Additional Results

Considering regression datasets from the UCI repository, we report the mean and standard deviation of the root mean squared error (RMSE) from 10-fold cross validation[4]. Also presented is the mean training time per fold on a machine with two E5-2680 v3 processors and 128Gb of RAM, and the expected sparsity (percentage of zeros) within a posterior sample. Using a generalized linear model, we consider $b = 2000$ random Fourier features of a squared-exponential kernel with automatic relevance determination [31]. Before generating the features, we initialize the kernel hyperparameters including the prior variance $\sigma_w^2$ and the Gaussian noise variance $\sigma^2$ by maximizing the marginal likelihood of an exact Gaussian process constructed on $\min(n, 1000)$ points randomly selected from the dataset [36]. All discretely relaxed models (containing "DIRECT"), only have support at $w \in \texttt{linspace}(-3\sigma_w, 3\sigma_w, \bar{m}=15)$, allowing $\mathbf{w}$ to be stored as 4-bit quantized integers.

For REPARAM we perform doubly stochastic optimization using a mini-batch size of 100 and using 10 Monte Carlo samples for the gradient estimates at each iteration. For datasets with $n < 3000$ we optimize for 1000 iterations and we optimize for 10000 iterations for all larger datasets. This model was implemented in Edward [37]. For the DIRECT mean-field model we use an L-BFGS optimizer [32] and run until convergence, or 1000 iterations are reached. For the DIRECT 5-mixture model we perform stochastic gradient descent using $t = 3000$ Monte Carlo samples for the entropy gradient estimator eq. (11).

|  |  |  | Discrete 4-bit Prior | | |
|  |  |  | DIRECT 5-Mixture ELBO-LB | | |
| Dataset | $n$ | $d$ | Time | RMSE | Sparsity |
|---|---|---|---|---|---|
| challenger | 23 | 4 | 15 | $0.528 \pm 0.243$ | 16% |
| fertility | 100 | 9 | 15 | $0.16 \pm 0.04$ | 16% |
| automobile | 159 | 25 | 24 | $0.137 \pm 0.053$ | 47% |
| servo | 167 | 4 | 24 | $0.282 \pm 0.067$ | 32% |
| cancer | 194 | 33 | 17 | $23.344 \pm 3.414$ | 18% |
| hardware | 209 | 7 | 24 | $0.492 \pm 0.117$ | 46% |
| yacht | 308 | 6 | 5 | $0.23 \pm 0.077$ | 96% |
| autompg | 392 | 7 | 24 | $2.624 \pm 0.339$ | 29% |
| housing | 506 | 13 | 24 | $2.782 \pm 0.324$ | 37% |
| forest | 517 | 12 | 15 | $1.361 \pm 0.159$ | 16% |
| stock | 536 | 11 | 233 | $0.011 \pm 0.002$ | 98% |
| pendulum | 630 | 9 | 6 | $1.36 \pm 0.227$ | 68% |
| energy | 768 | 8 | 5 | $3.116 \pm 0.218$ | 99% |
| concrete | 1030 | 8 | 19 | $5.571 \pm 0.665$ | 81% |
| solar | 1066 | 10 | 24 | $0.799 \pm 0.192$ | 22% |
| airfoil | 1503 | 5 | 16 | $2.175 \pm 0.32$ | 46% |
| wine | 1599 | 11 | 24 | $0.486 \pm 0.047$ | 50% |
| gas | 2565 | 128 | 5 | $0.204 \pm 0.053$ | 84% |
| skillcraft | 3338 | 19 | 78 | $0.253 \pm 0.017$ | 97% |
| sml | 4137 | 26 | 7 | $0.675 \pm 0.044$ | 57% |
| parkinsons | 5875 | 20 | 8 | $0.642 \pm 0.06$ | 13% |
| poletele | 15000 | 26 | 24 | $13.728 \pm 0.447$ | 16% |
| elevators | 16599 | 18 | 5 | $0.124 \pm 0.003$ | 99% |
| protein | 45730 | 9 | 16 | $0.62 \pm 0.007$ | 76% |
| kegg | 48827 | 20 | 6 | $0.222 \pm 0.01$ | 95% |
| ctslice | 53500 | 385 | 6 | $6.036 \pm 0.163$ | 19% |
| keggu | 63608 | 27 | 6 | $0.139 \pm 0.004$ | 87% |
| 3droad | 434874 | 3 | 14 | $10.487 \pm 0.075$ | 40% |
| song | 515345 | 90 | 8 | $0.502 \pm 0.002$ | 31% |
| buzz | 583250 | 77 | 9 | $1.009 \pm 0.004$ | 82% |
| electric | 2049280 | 11 | 5 | $0.593 \pm 0.036$ | 99.6% |

Table 2: Using a mixture variational distribution along with the the ELBO lower bound presented in Theorem 3, we present the mean and standard deviation of test error, average training time, and average expected sparsity of a posterior sample from 10-fold cross validation on UCI regression datasets.

For the DIRECT mean-field model we initialize the variational distribution to the prior. For the DIRECT mixture models, we first run the mean-field model and then initialize each mixture component to be randomly perturbed from the mean-field solution, and we initialize **a** to the mean-field solution. We initialize the mixture probabilities to be constant.

For predictive posterior mean computations, we use the exact computation presented in eq. (9) for both the DIRECT and mixture models. For REPARAM, we approximate the mean by sampling the variational distribution using 1000 samples.

In table 2 we consider again an unfactorized mixture variational distribution, however, we maximize the ELBO lower bound derived in Theorem 3. Since the ELBO gradients are deterministic, we again use an L-BFGS optimizer for training. In addition to the $150,000$ variational parameters used by the DIRECT 5-Mixture SGD model in table 1, computing the ELBO lower bound involves the simultaneous optimization of **a**, adding $30,000$ additional optimization parameters.