[Reviews · NeurIPS 2018]

Reviewer 1



Update: read the author feedback and all reviews and still agree the paper should be accepted. This paper addresses the problem of performing Bayesian inference on mobile hardware (e.g., self-driving car, phone) efficiently. As one would imagine, approaches that operate with discrete values have an advantage in hardware. Variational inference, a method for approximate Bayesian inference, often involves continuous latent variables and continuous variational parameters. This paper’s contribution is to cast everything in the discrete space with an approximating discrete prior. The authors explore and develop variational inference in such a setting and this appears to be a new idea in the field. I like the approach (called DIRECT) but a few things remain in my mind that were not addressed in the paper. First, let me state that I did not understand all the details of the technique. It was not clear to me how to choose the discretization granularity: with discrete approximations of continuous variables it usually matters how granular the points are. I also don’t understand what exactly DIRECT gives up with the Kronecker matrix representation. It’s clear that it is impossible to operate in high dimensional discrete space so there is probably a smoothness assumption embedded in the representation but I do not have a good intuition about that. The paper is well written and I like the worked examples with GLMs and neural networks. However, I found the beginning of Section 3 was hard to follow because of notation and the way the matrix W is introduced. Organizationally, the section on “predictive posterior computations” seems like it’s in the middle of the two worked examples. The experimental section was very thorough. With regards to related work, it would also be nice to compare to a more recent method than REINFORCE, which is known to have high variance. Also, whenever it is claimed that you compute the exact ELBO it needs to be clarified that it is only exact *after* the discrete approximation is applied. One minor addendum: a small niggle throughout the paper was claiming that you calculate “unbiased, zero-variance gradient estimators”. It’s not wrong but it just sounds like a roundabout way of saying “exact gradients”.

Reviewer 2



After reading the rebuttal and other reviews, I am convinced that this paper is a welcome contribution to NIPS. Thank you for sharing this idea! ------- The authors propose DIRECT, a method which computes the variational objective for learning discrete latent variable models exactly and efficiently according to Kronecker matrix algebra. They demonstrate accurate inference on latent variables discretized up to 4-bit values, and train on millions of data points. The paper is well-written and concise. I grealy enjoyed reading the motivation, main ideas, and experiments. The core idea comes from Proposition 3.1, in which inner products of Kronecker product vectors can be computed in linear rather than exponential time (i.e., exponential in the number of discrete latent variables b according to the number of possible outcomes they take m). By utilizing probabilistic models and variational approximations where ELBO optimization only results in inner products of this form, they can perform optimization of the exact ELBO. It would be great if the authors could comment on how this might extend to hybrid cases. For example, certain variables may be correlated (in the prior or variational posterior) and in those cases, it may be okay to spend the exponential complexity or even use Monte Carlo sampling. I enjoyed the derivations for the GLM as well as preliminary experiments on discretely relaxing Gaussian latent variables. Note the abstract, intro, and conclusion greatly overclaim significance. The idea is restricted to a specific class of models that factorizes across dimensions in both probability model and variational approximation. This is far less general than REINFORCE and it would be useful if the authors are clearer about this.

Reviewer 3



UPDATE: Read the review, still think this is a good paper. Score remains unchanged. The paper proposes a discretization scheme for variational inference with continuous latent variables. ELBO computations are reduced to summations, as distributions become high-dimensional tensors. While the approach may sound questionable at first due to the curse of dimensionality, the authors convincingly demonstrate that all relevant objects in mean-field variational inference appear to be low-rank tensors, whose inner products can be carried out in linear time of the latent variable dimension (as opposed to exponential, when done naively). Furthermore, the authors show that data-dependent terms can be efficiently precomputed, such that the resulting optimization problem scales independently with the number of data points. I found the results of the paper very surprising and interesting. I have two comments. First, the notion of a discrete relaxation is confusing; a more natural terminology would be a discretization. Second, it is certainly misleading to claim that the resulting gradient estimators are unbiased. Maybe the authors should stress that there is still a bias in the approach. I could not identify any obvious flaws in the paper, although I did not check the math in great depth. This is potentially a very interesting new research direction, which is why I consider the paper a significant new contribution.